# A climate adaptation strategy for Mai Po Inner Deep Bay Ramsar site: Steppingstone to climate proofing the East Asian-Australasian Flyway

Eric Wikramanayake[1]*, Carmen Or[1], Felipe Costa[2], Xianji Wen[1], Fion Cheung[1], Aurélie Shapiro[2,3]

1 WWF Hong-Kong, Hong Kong, Hong Kong SAR, 2 WWF-Germany Space+Science, Berlin, Germany, 3 Here+There Mapping Solutions, Berlin, Germany

* ericw@wwf.org.hk

**Data Availability Statement:** The land use data in Fig 1 are prepared by WWF Hong Kong, and was also used to derive the model outputs in Figs 2 and 3. The data are available here: https://globil-panda.

## Abstract

The ecological functionality of the East Asian-Australasian Flyway is threatened by the loss of wetlands which provide staging and wintering sites for migrating waterbirds. The disappearance of wetland ecosystems due to coastal development prevents birds from completing their migrations, resulting in population declines, and even an eventual collapse of the migration phenomenon. Coastal wetlands are also under threat from global climate change and its consequences, notably sea level rise (SLR), extreme storm events, and accompanying wave and tidal surges. The impacts of SLR are compounded by coastal subsidence and decreasing sedimentation, which can result from coastal development. Thus, important wetlands along the flyway should be assessed for the impacts of climate change and coastal subsidence to plan and implement proactive climate adaptation strategies that include habitat migration and possibility of coastal squeeze. We modelled the impacts of climate change and decreasing sedimentation rates on important bird habitats in the Mai Po Inner Deep Bay Ramsar site to support a climate adaptation strategy that will continue to host migratory birds. Located in the Inner Deep Bay of the Pearl River estuary, Mai Po's tidal flats, coastal mangroves, marshes, and fishponds provide habitat for over 80,000 wintering and passage waterbirds. We applied the Sea Level Affecting Marshes Model (SLAMM) to simulate habitat conversion under two SLR scenarios (1.5m and 2.0m) for 2050, 2075, and 2100 for four accretion rates (2mm/yr, 4 mm/yr, 8 mm/yr, 15 mm/yr). The results showed no discernible impact to habitats until after 2075, but projections for 2100 show that the mangroves, marshes and tidal flats could be impacted in almost all scenarios of SLR and accretion. Under a 1.5m SLR scenario, even at low tide, if accretion levels decrease to 4 mm/yr, the tidal flats will be inundated and with a 2 mm/yr accretion the mangroves will also be inundated. Thus, important shorebird habitats will be lost. During high tide the ponds inside the nature reserve, which are intensively managed to provide high tide roosting sites and other habitats for waterbirds, will also be inundated. Thus, with a 1.5m SLR and declining sedimentation the migratory shorebirds will lose habitat, including the high tide roosting habitats inside the nature reserve. The model also indicates that the fishponds further inland in the

opendata.arcgis.com/datasets/mai-po-wetland-habitats with a Creative Commons Attribution 4.0 International License, and copyright is duly referenced in the manuscript. The underlying SLAMM datalayers have been uploaded to Zenodo, and are available here: https://doi.org/10.5281/zenodo.3935747.

**Funding:** The analysis was supported by the general Mai Po Nature Reserve management fund, which is capitalized from multiple sources and managed by WWF Hong Kong. The salaries of EW, CO, XW, FC are supported by WWF Hong Kong (but not from the Mai Po Nature Reserve management fun). Authors FC and AS were provided with a consultancy by WWF HK, through Here+There Mapping Solutions to conduct the analysis. The funder (WWF Hong Kong) provided support in the form of salaries for authors [EW, CO, XW, FC], but did not have any additional role in the study design, data collection and analysis, decision to publish, or preparation of the manuscript. The specific roles of these authors are articulated in the 'author contributions' section. FC and AS received a consultancy from WWF Hong Kong through Here+There Mapping Solutions, to run the SLAMM model.

**Competing interests:** FC and AS received a consultancy from WWF Hong Kong through Here+There Mapping Solutions, to run the SLAMM model. This does not alter our adherence to PLOS ONE policies on sharing data and materials.

Ramsar site will be less impacted. Most fishponds are privately owned and could be developed in the future, including into high rise apartments; thus, securing them for conservation should be an important climate change adaptation strategy for Mai Po, since they provide essential habitats for birds under future climate change scenarios. But Mai Po is only one steppingstone along the EAAF, and hundreds of other wetlands are also threatened by encroaching infrastructure and climate change. Thus, similar analyses for the other wetlands are recommended to develop a flyway-wide climate-adaptation conservation strategy before available options become lost to wetland conversion.

## Introduction

Millions of birds undertake annual long-distance seasonal migrations along the East Asian-Australasian Flyway (EAAF), which stretches for over 13,000 km from the Arctic Circle to Australia and New Zealand [1]. The ecological functionality of this migration corridor depends on a network of coastal wetlands that represent steppingstones used by the birds as staging areas and over-wintering sites [2, 3]. Over the past several decades many of the wetlands along the EAAF have been converted for development, from agriculture and aquaculture to coastal infrastructure [4–7]. Over 65% of the intertidal mudflats in the Yellow Sea, one of the more important staging areas along the flyway, have been lost in recent decades [8–10]. Continued loss of strategically located wetlands could drastically impact connectivity, preventing birds from completing their migrations, resulting in eventual species population declines, extirpation and eventual collapse of this important ecological and evolutionary phenomenon that has evolved over millennia [2, 3, 11].

But like many coastal wetlands across the world the EAAF wetlands are also under additional threat from global climate change and its consequences, notably sea level rise (SLR) and extreme storms that are accompanied by stronger wave and tidal surges [5, 12]. The impacts of SLR and natural disasters are further compounded by subsidence and decreasing coastal sedimentation that is necessary to counter SLR [13–16]. Iwamura et al. [17] estimate that between 23 to 40% of the intertidal habitat will be lost and unavailable for migratory birds along the EAAF because of sea level rise, with some regions losing more than others. Coastal development, which has resulted in loss of wetlands, can now also place the remaining wetlands in a situation where they will be subjected to coastal squeeze; unable to migrate further inland in the event of SLR because of the adjacent hard infrastructure [18–20]. Thus, important wetlands along the flyway should be assessed for the impacts of climate change and associated drivers of ecological and environmental change to plan and implement proactive climate adaptation strategies. These analyses should be conducted at site scales to identify site-scale conservation strategies based on habitat availability, current and planned landuse, and restoration opportunities. Because wetland conversion and coastal development is still continuing apace these analyses are urgently needed before options for conservation and restoration disappear.

In Hong Kong, projections suggest rising sea levels and warmer, wetter weather that can spawn a higher number of severe typhoons [21, 22]. But, while the sea is rising, the deltas and mudflats in the Pearl River Delta could be shrinking. A model of sediment deposition and coastal nourishment of the Pearl River Delta projected that the fluvial sediment supply that fed the Pearl River Delta has decreased by 71% from the previous 'natural state', and any potential compensation from increased river flows attributed to climate change will only be about 1% of this deficit by the end of the 21st century, assuming impacts from human activities are

maintained at current levels [23]. Under this scenario, the coastal zones will be starved of approximately 6,000 Mt of fluvial sediment until the end of this century, which may also result in the erosion and shrinking of the delta, including the inter-tidal mudflat [23]. In fact, recent studies of the sub-aqueous bathymetry of the Pearl River Delta has shown that sections of the delta are already eroding, or accreting at lower rates [24, 25].

The Mai Po Inner Deep Bay Ramsar site in Hong Kong is an important staging and wintering site along the EAAF and supports large numbers of migratory birds, including 24 globally threatened bird species associated with wetlands. Thus, rising sea levels, stronger storm surges and shrinking mudflats will combine to decrease the availability of habitats for the waterbirds in Mai Po in the future, and adaptation plans to counter the threats from climate change and offshore subsidence are needed now to ensure persistence of migrations and flyway functionality. Here, we modelled the impacts of climate change and decreasing sedimentation rates on the important bird habitats in the Mai Po Inner Deep Bay Ramsar site to support a climate adaptation strategy that will continue to host migratory waterbirds. To do this, we applied the SLAMM (Sea Level Affecting Marshes Model) version 6.7 [26] to simulate habitat conversion under selected sea-level rise scenarios for coastal areas in the Mai Po Inner Deep Bay Ramsar site and adjacent areas surrounding it.

This model has been widely used for coastal resource management across the world, including for selected wetlands in the Yellow Sea estuary of China [16, 27] and Korea [28] in the EAAF region. But we note that the lack of data on historic wetland conversion, especially outside North America [29] hinders retrospective analysis, which could be used to calibrate and validate the model. Analyses have shown that elevation data is the most important factor for output accuracy since conversion among habitat classes is mainly governed by elevation [29–31]. Therefore, a high-resolution DEM is important for generating consistent results. Other site-specific parameters such as tidal range and accretion also affect the model outputs and should therefore be as accurate as possible [30]. Despite these shortcomings, SLAMM is the only marsh migration model that handles uncertainty in spatial inputs (DEM, VDATUM, etc.) and parameter choices, and this capability has contributed to its wide use [31].

## Mai Po Inner Deep Bay Ramsar site

The Mai Po Inner Deep Bay Ramsar site (hereafter Mai Po) is located in the North West New Territories of Hong Kong Special Administrative Region (HKSAR), along the southern edges of the Deep Bay of the Pearl River estuary (Fig 1A). The Ramsar site is comprised of tidal mudflats, coastal mangroves, marshes, and fishponds (Fig 1B), and now represents Hong Kong's largest remaining wetland. It is about 1,500 ha in extent, with a 380 ha nature reserve zoned within it (Fig 1B).

Mai Po supports about 80,000 wintering waterbirds and another 20,000–30,000 use it as a staging area during their annual migration in spring and autumn. Regular visitors to Mai Po include the Endangered Black-faced Spoonbill (*Platalea minor*) and Nordmann's Greenshank (*Tringa guttifer*), and, occasionally, the Critically Endangered Spoon-billed sandpiper (*Calidris pygmaea*). Mai Po played a significant role in the population recovery of the Black-faced spoonbill, and now regularly supports about 7–10% of the global population [32].

In the 1950s, most of the mangroves in Mai Po and surrounding coastal areas were converted to a system of traditional aquaculture pond management known as *gei wai*, where the ponds were connected to the bay via a sluice gate that allows water exchange with the bay. Some mangroves were left to remain in the ponds, and the biomass from the mangroves nourished the shrimp, oysters and fishes that were stocked in the ponds through tidal water exchanges, which also aerated and regularly flushed the ponds. Shrimp were harvested in the

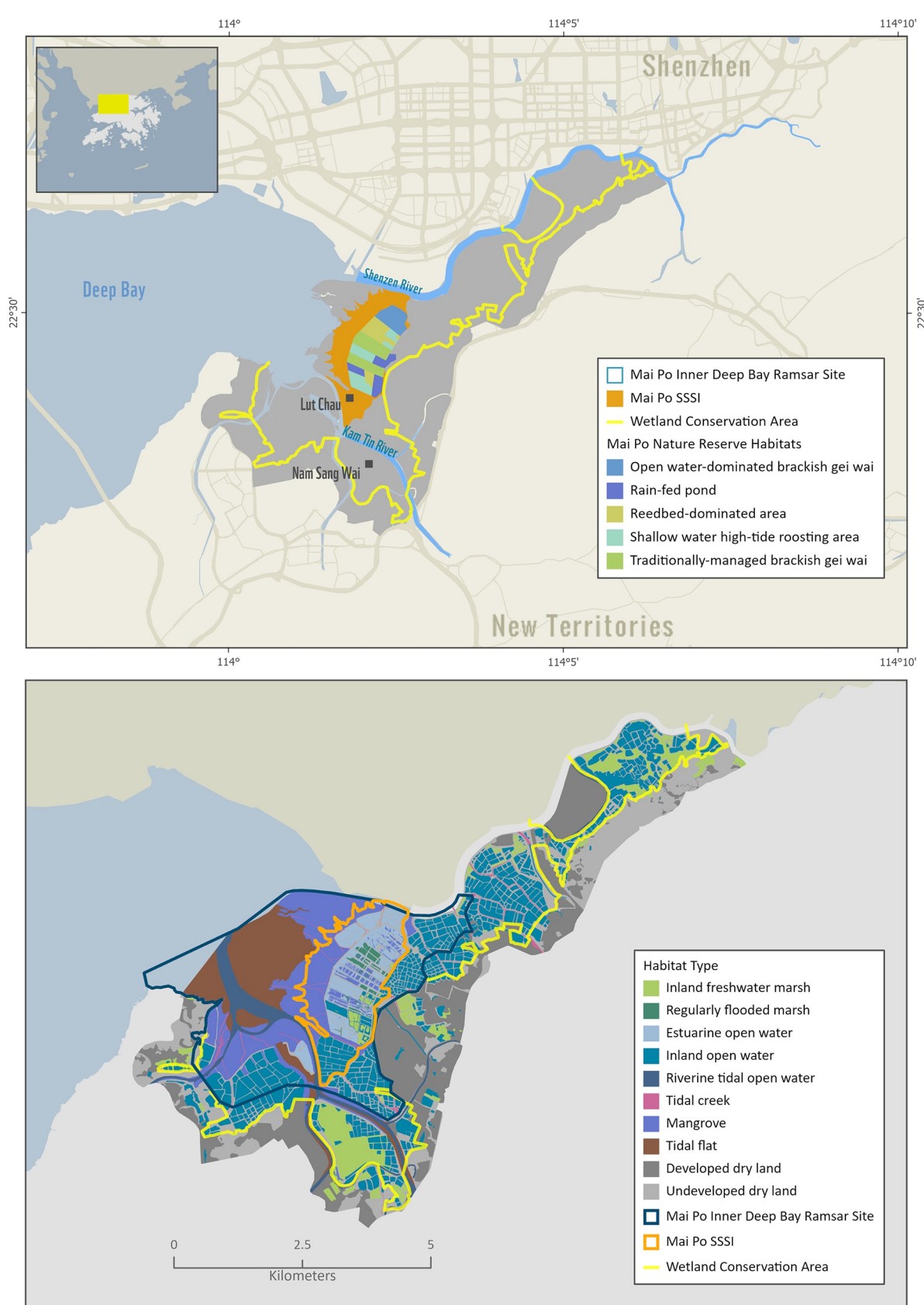

**Fig 1.** The location of the Mai Po Inner Deep Bay Ramsar site in relation to the Deep Bay (A, above) and major habitat types (B, below) that were cross-walked to the SLAMM. categories, as indicated in Table 1. The Shenzhen River borders the northern boundary of the Wetland Conservation Area. The Kam Tin River is south of the Mai Po SSSI. Basemap Sources: Esri, DeLorme, HERE. Landuse maps and boundary layers for the wetland conservation area, Ramsar site, and Nature Reserve are from WWF Hong Kong. https://globil-panda.opendata.arcgis.com/datasets/mai-po-wetland-habitats.

summer, and the *gei wai* were drained during the Autumn-Winter for harvesting bigger fishes and maintenance. During this time the birds are able to use them as feeding areas. Thus, these *gei wai* are a good example of 'wise use' management of wetlands that supported both migratory waterbirds and local fisher livelihoods. Since the 1960's, however, most *gei wai* have been converted to non-traditional fishponds with no connections to the bay and used for commercial grow out of fishes, mostly exotic species. However, the *gei wai* in the Mai Po Nature Reserve are operated in the traditional way to maintain habitat for waterbirds.

The nature reserve is also part of the Site of Special Scientific Interest (SSSI), zoned because of its biological importance, and is comprised of seven Biodiversity Management Zones managed under controlled hydrological regimes to provide shallow and deep open water areas, reedbed dominated ponds, rain-fed ponds, and *gei wai* to provide the range of habitats and food for waterbirds and other birds that use the nature reserve. Several active, commercial fishponds surround the nature reserve, inside the landward areas of Mai Po and beyond, in an area designated as a Wetland Conservation Area (Fig 1B).

## Methods

We projected impacts for 3 time intervals, 2050, 2075, and 2100 in relation to 2019 for two different SLR scenarios, 1.5 m and 2.0 m until 2100, and for four accretion rates; i.e., 2mm/yr, 4 mm/yr, 8 mm/yr, 15 mm/yr. We used these accretion rates because historical records (between 1898 and 1949) estimate accretion at 8 mm/year, but the rate had increased to about 15 mm/yr in the 1980's and 1990's, attributed to land clearing for agriculture and development in Shenzhen [23, 33]. From the 1990's onwards, there has been a decrease in fluvial sediment supply to the coast, and models suggest that sediment loads are expected to decrease further, by about 70% in the future with sediment trapping behind the Yangtan and Longtan mega dams [23]. Therefore, we applied 2 and 4 mm/yr accretion rates, which are approximately 30% of the 8 and 15 mm/yr rates, assuming a proportionate deposition based on declining sediment loads.

Models of SLR [21] projected an average rise of about 0.67 m and 0.84 m for Representative Concentration Pathways (RCP) 4.5 and 8.5 respectively, in the seas off Hong Kong, which is 0.2m higher than the global mean values projected by the Fifth Assessment Report of Intergovernmental Panel on Climate Change [34]. Global temperature increases between 1.9 and 3.1˚C can bring about fast Antarctic disintegration that can contribute an additional 0.21 to 0.74 m to SLR this century within 5 and 95% quantiles, respectively, of the RCP 8.5, and contributions of up to 0.45 m for the 95% quantile under RCP 4.5 [35]. The temperature range includes the 2˚C target set by the Paris CoP. Thus, in this analysis we used 1.5 and 2 m SLR scenarios to approximate the 1.5 m (median) and 2 m (95% quantile) for RCP 8.5 [35], because experts believe that even the upper bound of the 'business as usual' RCP 8.5 trajectory could likely be exceeded as few countries are adhering to their Nationally Determined Contributions to mitigate global climate change [36]. This justifies the use of SLR scenarios that even exceed 2 m SLR for future casting and planning [37].

To execute the SLAMM, we first reclassified the habitat and land use-land cover in the Ramsar site and the Wetland Conservation Area (Fig 1) into corresponding SLAMM categories (Table 1). We used expert opinion based on our experience in the Mai Po area and ecological definitions of the SLAMM categories in Clough et al. [26] to do the cross-walk. We used a

**Table 1. Land use and habitat categories in Mai Po Inner Deep Bay Ramsar site and Wetland Conservation Area cross-walked to SLAMM categories (based on SLAMM 6.7 technical documentation, Clough et al. [26]).**

| Land use and habitat classes in Mai Po | SLAMM categories | SLAMM Code |
|---|---|---|
| Urban; Buildings; Built up areas | Developed Dry Land | 1 |
| Bare soil; Pond bunds; Gei wai/pond islands in nature reserve; Mixed vegetation; Sparse tree/forest | Undeveloped Dry Land | 2 |
| Marshes (inland and outside the Ramsar site); Reedbeds outside nature reserve and pond 23/24 in nature reserve | Inland-Fresh Marsh | 5 |
| Reedbed and emergent in nature reserve ponds | Regularly-Flooded Marsh | 8 |
| Mangrove | Mangrove | 9 |
| Intertidal mudflats; Beach, Tidal Marsh (along Kam Tin River) | Tidal Flat | 11 |
| Freshwater ponds in nature reserve; Lily ponds in nature reserve; Fishponds inside Wetland Conservation Area | Inland Open Water | 15 |
| Tidal channel (through tidal flat), River | Riverine Tidal | 16 |
| Ponds adjacent to the Shenzhen River; Open water areas in nature reserve ponds | Estuarine Open Water | 17 |
| Open water and riparian vegetation along tidal Creeks | Tidal Creek | 18 |

5m Digital Elevation Model for Hong Kong to calculate percent slope using the function in ESRI ArcMap 10.7. These three layers (i.e., land cover, DEM, and slope) were used as the main input files for the simulations. Since there was no information on dikes this option was not applied, along with the soil saturation parameter. We also used the following input parameters: offshore direction is west; MTL-NAVD88 (m) = 0 and 0.7m, for low and high tide, respectively; GT Great Diurnal Tide Range (m) = 1.46 m, based on information from the Hong Kong Observatory data (https://www.hko.gov.hk/en/tide/predtide.htm?s=TBT).

In this analysis, low and high tides are used to denominate different modelling scenarios. Since the average tidal range for the site is 1.46 m, we applied two scenarios of simulation; low and high tide. For the low tide scenario, we simulated no height difference between elevation data and local datum. For the high tide scenario, we applied 0.7 m of height difference, which represents half of the total tidal range. The objective was to derive two results to define a range (i.e. minimum and maximum), since the exact delineation of the shoreline is unknown.

The choice for 0.7 m was made after a sensitivity analysis over the "time-zero" run of SLAMM to calibrate the model to site-specific data. We then tested the model with increasing heights until we perceived changes in the classes and selected 0.7 m as the upper boundary, therefore called high-tide scenario.

## Results

There was no discernible impact to the habitats in Mai Po until after 2075, and we do not present the results for the earlier timelines here. However, projections for 2100 suggest that overall, the mangroves, marshes and tidal flats are impacted in almost all scenarios of SLR and accretion (Table 2). Under a 1.5 m SLR scenario, at low tide there was little change in the tidal mudflats and coastal mangroves if the current accretion levels of 8 mm/yr or more continues into the future (Table 2, Fig 2). However, if accretion levels decrease, as predicted, to 4 mm/yr, the tidal flats are expected to be inundated even during low tide with over 77% being lost as habitat for wading and shorebirds, while the mangroves remain relatively unaffected (Table 2, Fig 2). When accretion levels decline to 2 mm/yr, over half the coastal mangroves could also be lost due to permanent inundation of the coastal areas, including during low tide (Table 2, Fig 2). A

**Table 2. Changes to habitats under the 1.5 and 2 m sea level rise scenarios and different accretion rates by 2100 as projected by the SLAMM.**

| | 1.5 m Sea Level Rise | | 2 m Sea Level Rise | |
|---|---|---|---|---|
| | Low Tide | High Tide | Low Tide | High Tide |
| | %Change in Habitat | | %Change in Habitat | |
| | 2100 | 2100 | 2100 | 2100 |
| **2 mm Accretion** | | | | |
| Regularly-Flooded Marsh | -56 | -57 | -57 | -80 |
| Mangrove | -54 | -63 | -58 | -61 |
| Tidal Flat | -77 | -91 | -77 | -92 |
| Inland Open Water/ Inland Fresh Marsh | -18 | -22 | -19 | -59 |
| **4mm Accretion** | | | | |
| Regularly-Flooded Marsh | -1 | -57 | -57 | -79 |
| Mangrove | 3 | -62 | -57 | -61 |
| Tidal Flat | -77 | -77 | -77 | -92 |
| Inland Open Water/ Inland Fresh Marsh | -16 | -22 | -19 | -58 |
| **8 mm Accretion** | | | | |
| Regularly-Flooded Marsh | 0 | -57 | -56 | -57 |
| Mangrove | 3 | -62 | -57 | -53 |
| Tidal Flat | 0 | -77 | -77 | -91 |
| Inland Open Water/ Inland Fresh Marsh | -15 | -22 | -18 | -46 |
| **15 mm Accretion** | | | | |
| Regularly-Flooded Marsh | 0 | 0 | 0 | -57 |
| Mangrove | 3 | 0 | 0 | -49 |
| Tidal Flat | 0 | -76 | 0 | -77 |
| Inland Open Water/ Inland Fresh Marsh | -15 | -19 | -16 | -45 |

Percent losses are given for low and high tide scenarios.

1.5 m SLR will also result in a loss of 15 to 18% of inland and open water marshes in the northern reaches of the wetland conservation area (Table 2, Fig 2). These marshes represent the fishponds, and the area bordering the lower reaches of the Shenzhen River and upper estuary, where the river bank is artificially constructed for river training, and there are no coastal mangroves or tidal mudflats. The regularly flooded marshes represent the ponds that support reedbeds in the Nature Reserve, and under a 2 mm/yr accretion rate, over 50% of these marshes will be lost, indicating that some of the ponds that are not connected to the estuary via tidal creeks will also become inundated during low tide under a 1.5 m SLR (Table 2). This likely reflects erosion and subsidence of the tidal flats, allowing sea water intrusion into the nature reserve.

Under a more extreme 2 m SLR scenario, the tidal flats and coastal mangroves will still remain relatively unaffected during low tide at accretion rates of 15 mm/yr, but there will be a loss of about 77 and 57%, respectively, of tidal flats and mangroves with an 8 mm/yr accretion rate (Table 2). The regularly flooded marshes in the nature reserve will also lose 56% of area with an 8 mm/yr accretion rate, and the inland open water marshes will lose about 18% of area even with a 15 mm/yr accretion rate.

Under the high tide scenarios, with 1.5 m SLR and 15 mm accretion levels about 76% of the tidal flats will become inundated, while the mangroves will remain unaffected (Table 2, Fig 3). There will be a loss of between 19 to 22% of inland open water marshes during high tide, a marginal increase from the low tide scenario (Table 2). However, over half of the regularly

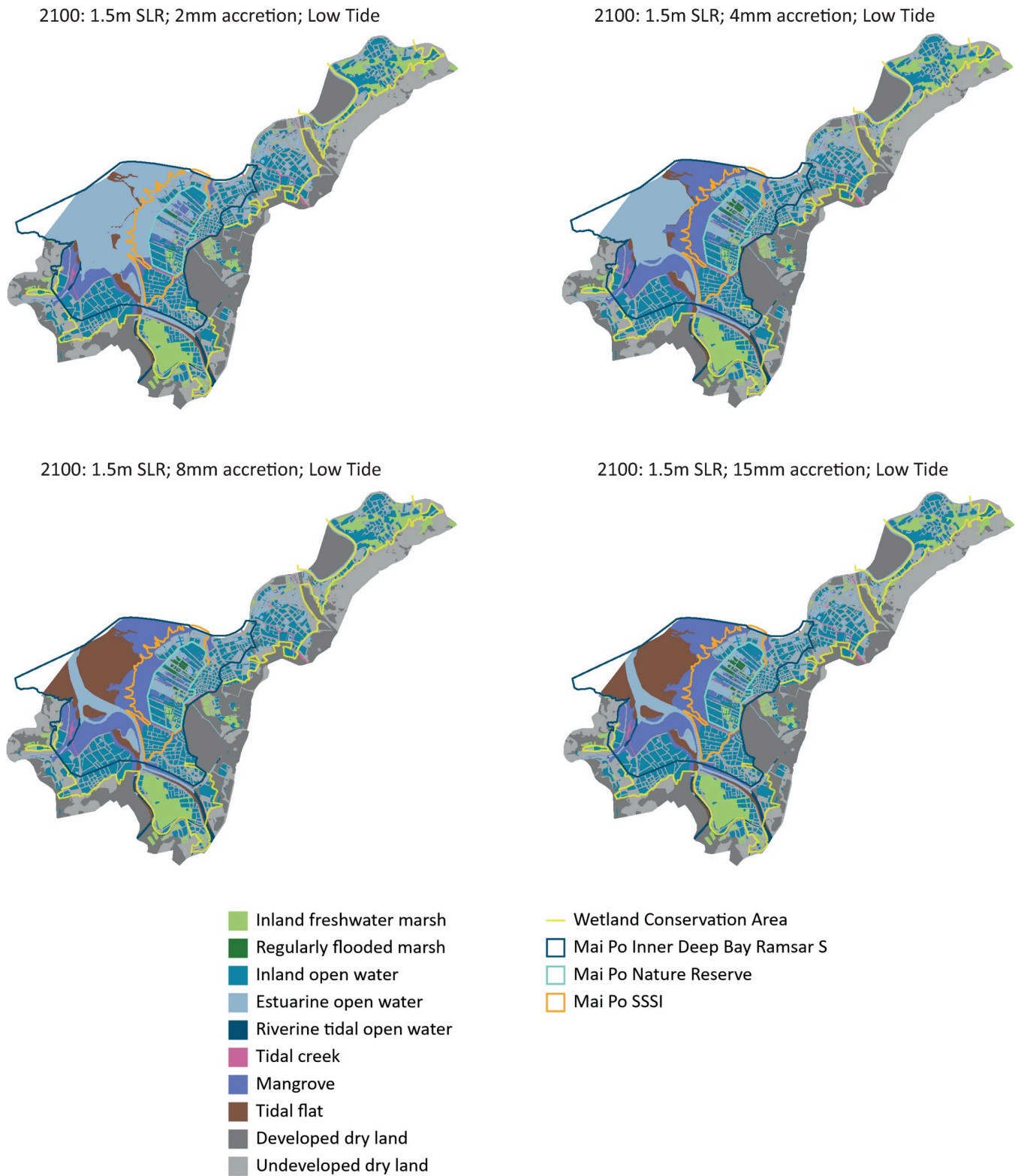

**Fig 2. SLAMM outputs derived from this analysis to show changes to habitats at low tide under 1.5 m sea level rise and 15mm, 8mm, 4mm and 2mm/ year accretion rates by 2100.** Landuse basemaps used in the analysis are from WWF Hong Kong https://globil-panda.opendata.arcgis.com/datasets/mai-po-wetland-habitats.

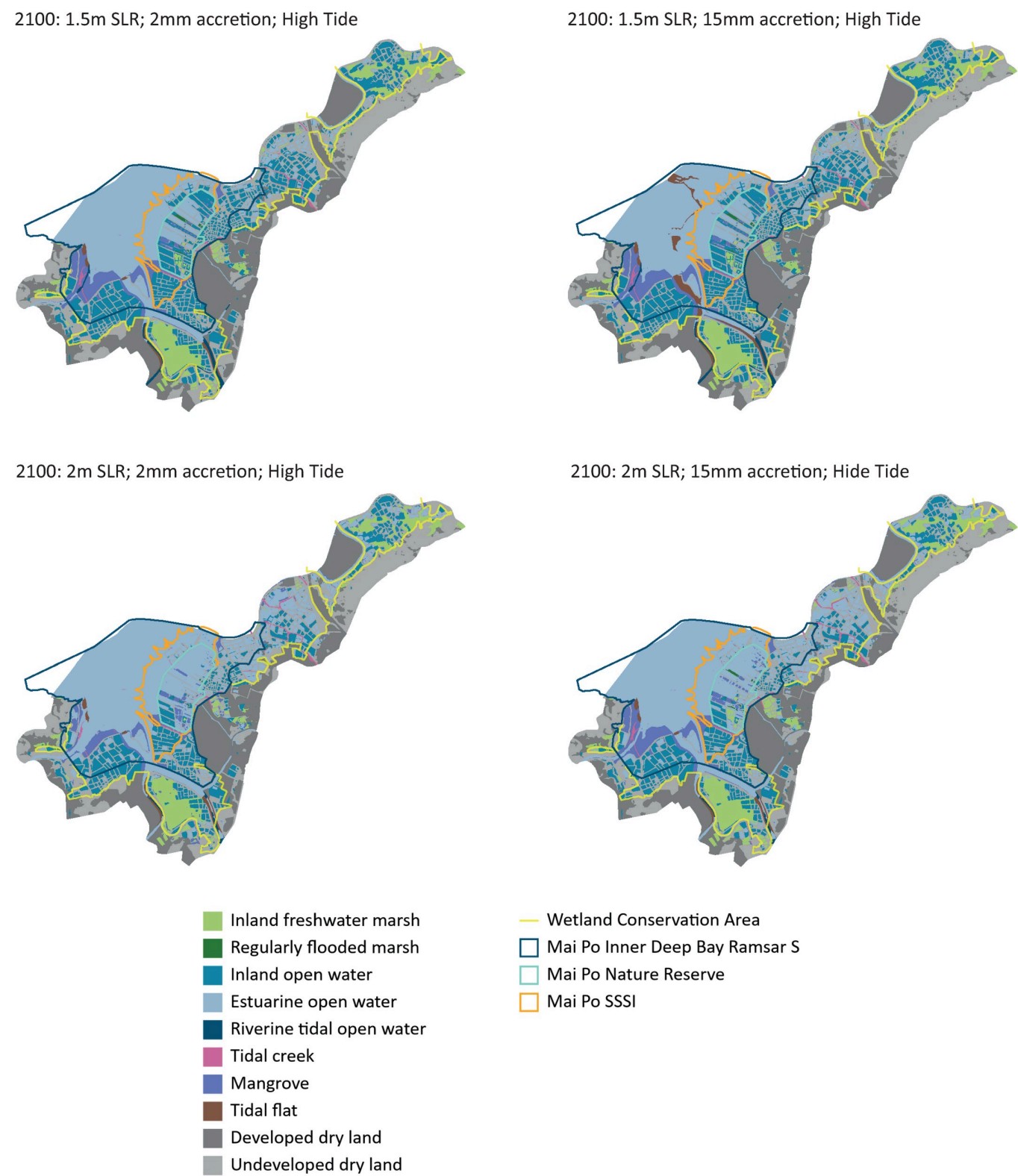

**Fig 3. SLAMM outputs derived from this analysis to show changes to habitats at high tide under 1.5 and 2 m sea level rise and 15mm and 2mm/year accretion rates by 2100.** Landuse basemaps used in the analysis are from WWF Hong Kong https://globil-panda.opendata.arcgis.com/datasets/mai-po-wetland-habitats.

flooded marshes inside the nature reserve will also be inundated even with an 8 mm/yr accretion rate and less (Table 2). Under a 2 m SLR scenario, during high tide over 77% of the tidal flats and about half of the mangroves and marshes will be inundated even with a 15 mm accretion rate (Table 2). If the accretion decreases to 4 mm/yr or less, most of the marshes, mangroves, and tidal flats will be underwater.

## Discussion

Long distance animal migrations are some of Nature's more wondrous phenomenon. But across the world these migrations are threatened by habitat loss, barriers that constrain or prevent movement, hunting [38], and now, climate change [11]. The EAAF is one of nine major long-distance corridors for migratory waterbirds that covers a swath over East, Southeast and eastern parts of South Asia to include 22 countries from Arctic Russia and USA (Alaska) to Australia and New Zealand [39]. The wetlands that lie scattered through this vast landscape, especially along the coastal areas, represent steppingstones where the waterbirds, which comprise a large group of the migrants, can rest, refuel, and continue on their journey, or provide winter habitats [40].

But the ecological functionality of the flyways that have evolved over millennia are now threatened [6, 8, 9, 41, 42]. Asia's coastal areas are among the most densely populated areas, and urban and industrial infrastructure is rapidly expanding at the expense of the natural ecosystems disregarding the importance of the ecosystem services these wetlands provide and the longer term economic and social consequences of losing them [5, 8, 43–46]. Over 80% of East and Southeast Asia's wetlands are now classified as threatened and nearly half of all tidal flats have already been lost [4]. Thus, many of the migratory waterbirds that depend on the wetlands along the EAAF are now facing population declines as important and strategic stopover and winter sites continue to disappear [1, 4, 8, 35]. While over 5 million individuals from 58 species of shorebirds are known to use the EAAF, it now has the highest number of threatened species and declining populations [47]. Analyses have shown that less than 10% of the wetlands that support waterbirds are protected, and many important yet unprotected areas are vulnerable to anthropogenic threats [3, 34, 48]. Thus, given the rapid loss of wetlands, inadequate protection and conservation management and consequent erosion of ecological connectivity and functionality of the flyway, it is important to secure the remaining wetlands to ensure that the migrations persist [35].

In addition to protection to prevent anthropogenic conversion, the wetlands and the flyway also require a 'climate-smart' conservation paradigm [49–51]. As transitional ecosystems between terrestrial and marine systems, coastal wetlands are vulnerable to SLR and the severe storms and associated strong wave surges expected from climate change on one side, and continued land conversion in the terrestrial realm [52, 53]. With SLR, coastal wetlands could migrate further inland [48, 54]. But if the adjacent terrestrial areas are converted to anthropogenic land uses this migration will be constrained and the wetlands will be lost due to coastal squeeze [48, 55]. As strategic wetlands along the flyway are lost to development and climate change and consequent environmental impacts, the functionality of the flyway will become degraded, and the migration phenomenon could eventually cease.

### Climate smarting the wetlands: The case of the Mai Po Inner Deep Bay Ramsar site

As an important steppingstone and wintering wetland for waterbirds along the EAAF, Mai Po represents a case study of how a climate adaptation strategy can be integrated into conservation management. The results from our climate model shows that over 230 ha of tidal flats,

which are critically important habitats, could become unavailable by the end of the century even under the more conservative estimate of 1.5 m SLR if accretion levels drop to below 8 mm/yr. This will greatly reduce the capacity of the Ramsar site to support the numbers of waterbirds that currently use it. A lower accretion rate will inundate the mangroves, and estuarine water intrusion could extend into the ponds inside the nature reserve. At high tide there will be more extensive inundation of the nature reserve and some of the marshes in the northern extents of the Wetland Conservation Area (Fig 3). These marshes—categorized as such for the SLAMM—are represented by fishponds. An analysis that applied SLAMM to coastal wetlands in Korea also showed similar results, where the tidal flats were lost to rising sea levels, with inundation of adjacent rice fields [28]. In China's Yangtze River estuary, models predict that accelerating sea level rise, land subsidence and low sediment accretion could result in considerable or complete loss of coastal wetland habitats [16]. Thus, the climate change scenario in Mai Po is consistent with the dynamics in other coastal wetlands along the flyway.

The *gei wai* and ponds inside the Mai Po Nature Reserve, which forms the core zone of the Ramsar site, are intensively managed by WWF Hong Kong for about 40 years, under contract from the Agriculture, Fisheries and Conservation Department (AFCD) of the HKSAR Government, according to 5-year prescriptive management plans. However, the tidal flats and the coastal mangroves remain largely unmanaged, with the exception of a 45 ha area in front of four floating bird watching hides to remove invasive sedges, grasses and mangrove seedlings that encroach into the mudflat. Management of the ponds in the nature reserve include maintaining islands, draining the ponds to re-contour the bottom to maintain the diversity of deep and shallow water habitats to support the assemblage of birds that require different habitats, and adapting the open areas to accommodate different species based on analysis of pond use by birds. While deeper ponds are maintained for the ducks and grebes, shallow water ponds and islands are maintained as high tide roosting sites for the shorebirds and waders when the tidal flats are inundated. Although these habitats are relatively well protected from direct anthropogenic land conversion due to the protected areas status of the nature reserve, the SLAMM outputs suggests that these habitats are vulnerable to the manifestations of global climate change. If SLR and declining sedimentation results in inundation of the tidal flats, coastal mangroves, and ponds in the nature reserve, the birds will lose habitat.

Because of sediment trapping by two large dams upstream in the Pearl River, there is expected to be a >70% decrease in fluvial sediment supply, and the coastal zone will be starved sediment deposition causing the Pearl River Delta to erode and shrink [23–25]. Thus, there is a high likelihood that the tidal flat will also erode, and happen even faster, and large areas of important habitat for the wading and shorebirds will be lost sooner than expected.

But the climate model suggests that the fishponds along the inland boundary and the southern areas of the Ramsar site, especially in Lut Chau and Nam Sang Wai, will be relatively less impacted by SLR (Figs 2 and 3). A few of the southern fishponds are outside the Ramsar site boundary but are still within the Wetland Conservation Area. Securing these fishponds for conservation should be a climate change adaptation strategy for Mai Po, since they could become wetland habitats for birds under future climate change scenarios, especially serving as high tide roosting sites when the ponds in the nature reserve are inundated. Artificial wetlands such as fishponds already serve as supratidal habitats for waterbirds during high tide when tidal flats are inundated even in other stopover sites and wintering areas across the flyway [56, 57]. But, while some of the wetlands are owned by the HKSAR Government and under the management jurisdiction of the AFCD, most wetlands in the Wetland Conservation Area are privately owned, including by property developers who have leased them to aquaculturists to manage and operate as commercial fishponds. Under HKSAR Government regulations, the wetlands and fishponds in the Wetland Conservation Area are protected but can be converted

based on the approval of the Town Planning Board. Thus, there is an imminent danger of the fishponds being converted into hard infrastructure, including high rise buildings. Such development will result in loss of potential habitat and conservation opportunities, and also create hard, impervious surfaces that effectively cause flooding elsewhere, and will be more difficult and economically unviable to restore back to wetlands in the future. An analysis of coastal squeeze in the Yellow River Delta has shown that farmlands, including those claimed from wetlands, allowed the potential for wetland migration and restoration because they are more pervious, relative to wetlands that had been converted to hard infrastructure [58].

## Ecosystems as climate proofing solutions for Hong Kong and the Greater Bay Area

Coastal wetlands are among the most productive and important ecosystems in the world [59]. In addition to supporting high biodiversity, they also provide a variety of ecosystem goods and services, including reducing flooding, protecting shorelines from wave and tidal surges expected from the more severe weather events due to climate change [3, 6, 14]. Thus, climate-smart land use strategies that prevent the conversion of wetlands can increase coastal resilience and diminish the vulnerability of coastal communities and infrastructure from the consequences of SLR and storms that make landfall [6, 60, 61].

According to the ADB [62] the economic losses from floods in the Asia-Pacific region could increase from the $6 billion per year in 2005 to $52 billion by 2050, and 13 of the top 20 cities with the largest predicted increase in annual losses will be in the Greater Bay Area. This area is also recognized as the main landfall area of tropical cyclones and typhoons from the Northwest Pacific and the South China Sea, with the typhoons being stronger and more intense, and accompanied by stronger storm surges [63]. Records from 1991 and 2005 show that storm surges, in effect, created a SLR of between 1.9 and 2.6 m in the Pearl River Delta [64], while severe typhoons can create surges that are as high as 4 m [22]. Thus, much of the Pearl River Delta and Greater Bay Area could be vulnerable to extreme weather events exacerbated by climate change, including SLR, floods caused by wave surges, saltwater intrusion, and an increase in the severity of storms [21, 22].

Coastal wetlands also buffer and attenuate storm surges and reduce property damage and loss of lives along the coastlines [65–69]. However, rapid economic development and population growth over the past four decades have resulted in widespread conversion and fragmentation of natural coastal habitats in the Greater Bay Area, making it more prone and vulnerable to these natural disasters, especially floods [70]. Removal of infrastructure to restore wetlands as climate buffers will be very expensive [71, 72]. Thus, forward-looking, proactive land-use planning for coastal areas that include climate impact analyses and nature-based solutions will be far more cost effective; the costs of failing to do so could be greater [14, 44, 73–77]. Conserving the existing coastal wetlands and preventing further conversion as an integrated climate response is an obvious choice.

## Climate proofing the flyway

The application of the SLAMM demonstrates the conservation opportunities for a climate adaptation strategy to ensure that Mai Po will continue to host migratory birds in the future. However, Mai Po is only one steppingstone along this vast corridor that is the EAAF, with hundreds of other such steppingstone wetlands that are also threatened by encroaching infrastructure and climate change [78]. The flyway needs all, or several strategically placed steppingstones to be functional to support the migration phenomenon [3, 43, 51]. Therefore, we recommend that this analysis be applied to the other wetlands that serve as strategic staging or

wintering sites along the EAAF—and other bird migration flyways—to develop landscape-scale climate adaptation strategies. Such plans will enable the migrations to persist, but also ease the vulnerabilities of people and economies concentrated in coastal regions. However, this should happen now, before available options become lost to the rapid and extensive conversion of wetlands from anthropogenic drivers and SLR from the landward and seaward sides, respectively.

## Acknowledgments

We thank the Peter Cornthwaite, CEO WWF Hong Kong and David Olson, Conservation Director for encouraging this study, Chi-Yeung Choi and a second anonymous reviewer for constructive comments. We also thank the Mai Po management team and management committee for their support.

## Author Contributions

**Conceptualization:** Eric Wikramanayake, Xianji Wen, Fion Cheung, Aurélie Shapiro.

**Data curation:** Eric Wikramanayake, Felipe Costa, Fion Cheung.

**Formal analysis:** Eric Wikramanayake, Carmen Or, Felipe Costa, Aurélie Shapiro.

**Investigation:** Eric Wikramanayake, Aurélie Shapiro.

**Methodology:** Eric Wikramanayake, Carmen Or, Felipe Costa, Xianji Wen, Aurélie Shapiro.

**Resources:** Xianji Wen, Fion Cheung, Aurélie Shapiro.

**Supervision:** Eric Wikramanayake, Aurélie Shapiro.

**Validation:** Fion Cheung.

**Writing – original draft:** Eric Wikramanayake.

**Writing – review & editing:** Eric Wikramanayake.

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
