## [Decision Letter · Decision Letter 0]

15 Jun 2020

PONE-D-20-12507

A Climate Adaptation Strategy for Mai Po Inner Deep Bay Ramsar Site: Steppingstone to Climate Proofing the East Asian-Australasian Flyway

PLOS ONE

Dear Dr. Eric Wikramanayake,

Thank you for submitting your manuscript to PLOS ONE. After careful consideration, we feel that it has merit but does not fully meet PLOS ONE’s publication criteria as it currently stands. Therefore, we invite you to submit a revised version of the manuscript that addresses the points raised during the review process.

We look forward to receiving your revised manuscript.

Kind regards,

Yong Zhang

Academic Editor

PLOS ONE

Additional Editor Comments:

In this manuscript, the authors modeled the change in coastal landscape in an important waterbird area (Mai Po) under different SLR and accretion scenarios. Both reviewers think this work is an important contribution in waterbirds conservation in eastern Asia. But they also raised some concerns such as the model limitations, constrains and accuracy. In addition, as suggested by one reviewer, it will be better to give some conservation advice in the discussion part. I also think the manuscript will be improved by taking these comments into account. Please see the detail comments below.

2. We note that Figures 1-3 in your submission contain map images which may be copyrighted.

a. You may seek permission from the original copyright holder of Figures 1-3 to publish the content specifically under the CC BY 4.0 license. 

'We thank the Peter Cornthwaite, CEO WWF Hong Kong and David Olson, Conservation Director for encouraging this study and providing the financial support'

'The funders had no role in study design, data collection and analysis, decision to publish, or preparation of the manuscript.'

'The authors have declared that no competing interests exist.'

We note that one or more of the authors are employed by a commercial company: Here+There Mapping Solutions

Reviewers' comments:

Reviewer's Responses to Questions

**Comments to the Author**

1. Is the manuscript technically sound, and do the data support the conclusions?

Reviewer #1: Partly

Reviewer #2: Yes

2. Has the statistical analysis been performed appropriately and rigorously? 

Reviewer #1: N/A

Reviewer #2: Yes

3. Have the authors made all data underlying the findings in their manuscript fully available?

Reviewer #1: Yes

Reviewer #2: Yes

4. Is the manuscript presented in an intelligible fashion and written in standard English?

Reviewer #1: Yes

Reviewer #2: Yes

5. Review Comments to the Author

Reviewer #1: The authors modelled the change in coastal landscape in an important waterbird area under different SLR and accretion scenarios. The authors then concluded that substantial amount of migratory bird habitat will be lost in 2100 under the predicted accretion rate and 1.5-2.0m SLR scenarios. This is study is critically important in conservation management and I added a few comments below which could be used to improve the manuscript.

General comments:

1. Given the modelling nature of this manuscript, please add a paragraph or section, to discuss briefly the limitations and constrains of this SLAMM approach. Has this model been tested? What was the accuracy? A brief discussion on these will allow less-experienced readers to evaluate the evidence and follow more easily.

2. Line 352 – this is an important paragraph but this also mean it's critically important to evaluate the model's accuracy, and the probability of errors – which could lead to the purchase of less useful area. Could the authors also demonstrate how much longer will other potential buffer area (ponds around Lut Chau and Nam Sang Wai) last for? If it's just 10-20 years, then is it really a sustainable solution?

3. From the waterbirds' perspective, e.g. shorebirds, this paper could be improved substantially by considering the entire Deep Bay instead of having an arbitrary cut off in the middle of the tidal flat. Many of the birds in Maipo are also known to the mainland China side of Deep Bay and leaving the other half out could only reveal partly how waterbird’s habitat will change in the future.

Specific comments:

Lines 27, 285, 393 – consider the use of “migration phenomenon” instead of “migration”

Line 119 – please provide the GPS coordinate, perhaps the centroid of Maipo

Line 133 – capitalize “E”ndangered

Line 136 – citation needed, consider the BFS annual census report or Sung et al. 2018

www.doi.org/10.1017/S0959270917000016

Line 165 – please provide a bit more details since accretion can still be clearly felt today, any information on when will the land clearing and development be stopped? How much of the accretion may come from natural runoffs?

Line 177 – the degree symbol looks more like a zero instead

Line 222 – Please explain what do low and high tides mean here (are they an average?), since substantial differences could be found between spring and neap tides

Line 252 – may also consider the latest publication related to hunting by Gallo-Cajiao et al. 2020 https://doi.org/10.1016/j.biocon.2020.108582

Line 339 – may also consider the latest publication by Jackson et al. 2020 https://doi.org/10.1016/j.biocon.2020.108591

Reviewer #2: The authors modelled the impacts of climate change and decreasing sedimentation rates on

important bird habitats in the Mai Po Inner Deep Bay Ramsar site to support a climate

adaptation strategy that will continue to host migratory birds. The text is grammatically under understandable, although English to be reviewed before considering it to publication. In addition, some of references are not in our style, which are close but not completely correct.

The work of this paper is practical and logical, it provides a great contribution to the literature on how the important bird habitats change in different future scenario. However, there are some main problems to be further improved as well, I listed some specific comments below:

As we know, the climate change and human disturbance are the main driving force of the coastal wetland change in the world. Base on the result, it seems the climate change is not the principal threat of the Mai Po till 2075. Hong Kong and Shenzhen both are the biggest city in the world. So, in the discussion, it needs to give more conservation advice or reflection, how to control or monitor the sediment. Such as, at the 2100: 2m SLR; 2mm accretion; High Tide scenario, it need build more high-tide habitat for waterbirds. Or we need to keep the sediment stabilize, try to avoid the construction and the activities of changing the ecological processes of wetland, e.g. Shenzhen Bay dredging for cruise recently.

It is better put the Low tide image and Hide tide of the same scenario together and it will be easy to check the impact for the migratory waterbird.

6. PLOS authors have the option to publish the peer review history of their article (what does this mean?). If published, this will include your full peer review and any attached files.

Reviewer #1: Yes: Chi-Yeung Choi

Reviewer #2: No

---

## [Author Response · Author response to Decision Letter 0]

22 Jul 2020

1. Map copyrights. Figure 1 has a basemap from ESRI. We have inserted the appropriate attribution in the figure caption (Basemap Sources: Esri, DeLorme, HERE) as per instructions here: https://support.esri.com/en/technical-article/000012040.

The landuse data in Figures 2 and 3 are prepared by WWF Hong Kong.

2. Acknowledgements Section: We have removed the reference to funding and funders in the acknowledgements statement. 

The analysis was supported by the general Mai Po Nature Reserve management fund, which is capitalized from multiple sources and managed by WWF Hong Kong. 

The salaries of EW, CO, XW, FC are supported by WWF Hong Kong (but not from the Mai Po Nature Reserve management fun). 

Authors FC and AS were provided with a consultancy by WWF HK, through Here+There Mapping Solutions to conduct the analysis.

Please do make the appropriate changes to the online submission form based on the above information, as indicated in your instructions (i.e., “Please include your amended statements within your cover letter; we will change the online submission form on your behalf.”)

3. Funding Statement and Competing Interests. Please change the Funding Statement and Competing Interests as follows: 

“The funder (WWF Hong Kong) provided support in the form of salaries for authors [EW, CO, XW, FC], but did not have any additional role in the study design, data collection and analysis, decision to publish, or preparation of the manuscript. The specific roles of these authors are articulated in the ‘author contributions’ section. FC and AS received a consultancy from WWF Hong Kong through Here+There Mapping Solutions, to run the SLAMM model. This does not alter our adherence to PLOS ONE policies on sharing data and materials.”

4. Data Availability statement. We will upload the underlying SLAMM datalayers to Zenodo

Reviewer’s Comments to the Authors

Reviewer #1: 

General comments:

1. Given the modelling nature of this manuscript, please add a paragraph or section, to discuss briefly the limitations and constrains of this SLAMM approach. Has this model been tested? What was the accuracy? A brief discussion on these will allow less-experienced readers to evaluate the evidence and follow more easily.

Author response: We have added some text to clarify the limitations and constraints of the SLAMM approach (lines 115-125). We also have text and provided references to show that the model has been applied to wetlands globally, including to wetlands in China and Korea, along the EAAF. We hope this will suffice.

2. Line 352 – this is an important paragraph but this also mean it's critically important to evaluate the model's accuracy, and the probability of errors – which could lead to the purchase of less useful area. Could the authors also demonstrate how much longer will other potential buffer area (ponds around Lut Chau and Nam Sang Wai) last for? If it's just 10-20 years, then is it really a sustainable solution?

Author response: Presumably this refers to the paragraph starting on line 331 (in the previous, uncorrected document). Only way to show that the wetlands will survive more than 10-20 years beyond 2100 would be to run the model for another 25 and/or 50 years. But increasing the time horizon will mean that the model outputs will also become less accurate and outputs less reliable.

However, we also note that our recommendations to conserving Lut Chau and Nam Sang Wai fishponds do not necessarily recommend purchasing these lands, but that they not be converted to hard infrastructure now or in the near future. They are currently maintained as fishponds, which are de-facto wetlands and are used by some waterbirds as supratidal wetlands. They can also be more easily and cost-effectively restored as wetlands for waterbirds as an adaptation strategy in the future, if not converted to hard infrastructure. Therefore, our proposal to maintain the wetlands is not on a timescale of 10-20 years beyond 2100, but starting now. 

3. From the waterbirds' perspective, e.g. shorebirds, this paper could be improved substantially by considering the entire Deep Bay instead of having an arbitrary cut off in the middle of the tidal flat. Many of the birds in Maipo are also known to the mainland China side of Deep Bay and leaving the other half out could only reveal partly how waterbird’s habitat will change in the future.

Author response: The reviewer is partly correct; the Deep Bay should be considered as an ecologically connected flyway site. However, while Mai Po and the mudflat area that was considered in the analysis is in Hong Kong SAR, the rest of the Deep Bay is under administration of the mainland China government, and we have been unable to acquire the required data layers to extend the analysis beyond Mai Po. 

Moreover, the objective of our analysis was to identify and recommend an adaptation strategy for the Mai Po Inner Deep Bay Ramsar site. Site-scaled analyses can be more relevant, sensitive, and contextual for site-based conservation management plans. However, we then go on to recommend that similar analyses be conducted for important sites along the EAAF that can be aggregated for a holistic, flyway-scaled adaptation strategy. 

Applying the model to larger scales will result in loss of site-specific sensitivity and cause important management recommendations to be glossed over. This has been a problem with many regional and global-scaled analyses that lose site-specific context. 

Specific comments:

Lines 27, 285, 393 – consider the use of “migration phenomenon” instead of “migration”

Author response: changes made.

Line 119 – please provide the GPS coordinate, perhaps the centroid of Maipo

Author response: The maps (Figure 1), which is meant to be a location map, provides the coordinates.

Line 133 – capitalize “E”ndangered

Author response: changes made.

Line 136 – citation needed, consider the BFS annual census report or Sung et al. 2018 www.doi.org/10.1017/S0959270917000016

Author response: reference added

Line 165 – please provide a bit more details since accretion can still be clearly felt today, any information on when will the land clearing and development be stopped? How much of the accretion may come from natural runoffs?

Author response: We have tried to clarify some of the time correlation between land-clearing and development trends and accretion, as requested by the reviewer as best as possible based on available published data. However, we also note that there is no reliable literature on current accretion rates. A monitoring study is now underway, but the data have not been published. We have thus provided the best available information and base our choice of parameters on these studies. As such, we have used—and presented—the range of possible accretion rate scenarios in our analyses. 

We were unable to find any reliable information on the contributions of natural runoff-based accretion (except the historical data that we use) and that attributed to land clearing. But we also note that such a breakdown is immaterial to our analysis, except to make landuse, land conversion, and restoration recommendations, which we do in a general way. We also note that this would be an important follow up study but would have to be undertaken in collaboration with researchers from mainland China since most of the upstream impacts from runoff originates there.

Line 177 – the degree symbol looks more like a zero instead

Author response: change made.

Line 222 – Please explain what do low and high tides mean here (are they an average?), since substantial differences could be found between spring and neap tides

Author response: text added (lines change made 208-217).

Line 252 – may also consider the latest publication related to hunting by Gallo-Cajiao et al. 2020 https://doi.org/10.1016/j.biocon.2020.108582

Author response: reference added

Line 339 – may also consider the latest publication by Jackson et al. 2020 https://doi.org/10.1016/j.biocon.2020.108591

Author response: reference added

Reviewer #2: 

As we know, the climate change and human disturbance are the main driving force of the coastal wetland change in the world. Base on the result, it seems the climate change is not the principal threat of the Mai Po till 2075. Hong Kong and Shenzhen both are the biggest city in the world. So, in the discussion, it needs to give more conservation advice or reflection, how to control or monitor the sediment. Such as, at the 2100: 2m SLR; 2mm accretion; High Tide scenario, it need build more high-tide habitat for waterbirds. Or we need to keep the sediment stabilize, try to avoid the construction and the activities of changing the ecological processes of wetland, e.g. Shenzhen Bay dredging for cruise recently.

Author response: We agree that in addition to climate change impacts there are very severe and more proximal direct and indirect anthropogenic threats to Mai Po and the Deep Bay. Some of these can be effectively mitigated, while others would require an adaptation strategy. What we have provided through this analysis is an adaptation strategy to the expected impacts of climate change. Monitoring and tackling the more proximal anthropogenic impacts, while also critically important, is beyond the scope of this paper. However, we have provided general recommendations on how nature-based solutions are required in the Greater Bay in an entire sub-section entitled “Ecosystems as climate proofing solutions for Hong Kong and the Greater Bay Area” in the Discussion, especially with respect to ‘climate proofing’ the bay. 

1. It is better put the Low tide image and Hide tide of the same scenario together and it will be easy to check the impact for the migratory waterbird.

Author response: We debated and thought through what the best way to group the maps would be, and eventually chose to group the low tide scenarios together to show that even during low tide, important bird habitats can be submerged under low sedimentation rate scenarios. And then to show the corresponding scenarios under high tide scenarios. Thus, while we concede that there are several ways to group these maps, we have chosen what we think is the best way to show what we wanted to show.

---

## [Decision Letter · Decision Letter 1]

16 Sep 2020

A Climate Adaptation Strategy for Mai Po Inner Deep Bay Ramsar Site: Steppingstone to Climate Proofing the East Asian-Australasian Flyway

PONE-D-20-12507R1

Dear Dr. Wikramanayake,

We’re pleased to inform you that your manuscript has been judged scientifically suitable for publication and will be formally accepted for publication once it meets all outstanding technical requirements.

Kind regards,

Yong Zhang

Academic Editor

PLOS ONE

Additional Editor Comments (optional):

I can only reach one of the original reviewers and he is happy with the revised version of the manuscript. I have reviewed the comments of the another reviewer and the responses from the authors, i think authors also answer the concerns appropriately.

Reviewers' comments:

Reviewer's Responses to Questions

**Comments to the Author**

1. If the authors have adequately addressed your comments raised in a previous round of review and you feel that this manuscript is now acceptable for publication, you may indicate that here to bypass the “Comments to the Author” section, enter your conflict of interest statement in the “Confidential to Editor” section, and submit your "Accept" recommendation.

Reviewer #1: All comments have been addressed

2. Is the manuscript technically sound, and do the data support the conclusions?

Reviewer #1: Yes

3. Has the statistical analysis been performed appropriately and rigorously? 

Reviewer #1: I Don't Know

4. Have the authors made all data underlying the findings in their manuscript fully available?

Reviewer #1: Yes

5. Is the manuscript presented in an intelligible fashion and written in standard English?

Reviewer #1: Yes

6. Review Comments to the Author

Reviewer #1: Well done in revising this interesting manuscript. All my concerns are addressed and let's hope the local stakeholders will follow the recommendations from this manuscript and secure those less-impacted wetlands for the long-term future.

7. PLOS authors have the option to publish the peer review history of their article (what does this mean?). If published, this will include your full peer review and any attached files.

Reviewer #1: **Yes: **Chi-Yeung Choi

---

## [Editor Report · Acceptance letter]

28 Sep 2020

PONE-D-20-12507R1 

A Climate Adaptation Strategy for Mai Po Inner Deep Bay Ramsar Site: Steppingstone to Climate Proofing the East Asian-Australasian Flyway 

Dear Dr. Wikramanayake:

I'm pleased to inform you that your manuscript has been deemed suitable for publication in PLOS ONE. Congratulations! Your manuscript is now with our production department. 

Kind regards, 

on behalf of

Dr. Yong Zhang 

Academic Editor

PLOS ONE